# The Relationships between Caregiver Burden, Physical Frailty, Race, Behavioral and Psychological Symptoms (BPSD), and Other Associated Variables: An Exploratory Study

**DOI:** 10.3390/medicina60030426

**Published:** 2024-03-01

**Authors:** Carl I. Cohen, Saeed Hashem, Kay Thwe Kyaw, Sharon A. Brangman, Suzanne Fields, Bruce R. Troen, Michael Reinhardt

**Affiliations:** 1Division of Geriatric Psychiatry, SUNY Downstate Health Sciences University, MSC 1203, 450 Clarkson Avenue, Brooklyn, NY 11203, USA; drkaykyaw007@outlook.com (K.T.K.); michael.reinhardt@downstate.edu (M.R.); 2McLean Hospital, 115 Mill St., Belmont, MA 02478, USA; sahashem@mclean.harvard.edu; 3Department of Geriatrics, SUNY Upstate Medical University, 175 Elizabeth Blackwell Street, Syracuse, NY 13210, USA; brangmas@upstate.edu; 4Division of General, Geriatric and Hospital Medicine, Renaissance School of Medicine, Stony Brook University, 101 Nicolls Road, Stony Brook, NY 11794, USA; suzanne.fields@stonybrookmedicine.edu; 5Division of Geriatrics and Landon Center on Aging, University of Kansas Medical Center and VA Kansas City Healthcare System, 4000 Cambridge Street, Kansas City, KS 66160, USA; btroen@kumc.edu

**Keywords:** dementia, frailty, caregiver burden, behavioral and psychological symptoms, neuropsychiatric symptoms, race

## Abstract

*Background and Objectives:* For persons with dementia, the relationships between caregiver burden, physical frailty, race, behavioral and psychological symptoms (BPSD), and other associated variables are poorly understood. Only one prior study examined the relationships among these variables but did not include race, which is an important social determinant of health outcomes in the United States. To examine these interactions, we conducted a cross-sectional exploratory study based on a model by Sugimoto and colleagues. *Materials and Methods:* The sample comprised 85 patient–caregiver dyads (58% White) seen in four centers in diverse regions of New York State. All patients met DSM5 criteria for a major neurocognitive disorder, had a Clinical Dementia Rating sum score of ≥3, and Mini-Mental State Examination (MMSE) score of 10 to 26. Other measures included the SHARE-Frailty Instrument(FI), the Neuropsychiatric Inventory (NPI) to assess BPSD, Zarit’s Caregiver Burden Interview (CBI), Lawton’s Activities of Daily Living (ADL) Scale, the MMSE, the Cumulative Illness Rating Scale for Geriatrics (CIRSG), age, and gender. *Results:* In our sample, 59% met the criteria for prefrail/subsyndromal or frail/syndromal (SSF) on the SHARE-FI. SSF had significant direct effects on the NPI and significant indirect effects on the CBI mediated through the NPI; the NPI had significant direct effects on the CBI. Race (White) had significant direct effects on the CBI (higher) and SSF (lower) but did not have significant indirect effects on the CBI. MMSE, ADL, and CIRSG were not significantly associated with the NPI or the CBI. *Conclusions:* Our analysis demonstrated that frailty, race, BPSD, and caregiver burden may directly or indirectly influence one another, and therefore should be considered essential elements of dementia assessment, care, and research. These results must be viewed as provisional and should be replicated longitudinally with larger samples.

## 1. Introduction

Dementia is thought to be one of the most serious public health challenges of the 21st century [1]. Not only does dementia affect the patient, but it can have a profound impact on caregivers. Perceived caregiver burden (CGB) among persons caring for dementia patients is associated with higher levels of stress, depression, and anxiety than in other caregivers of non-dementia patients [2] and is associated with severe adverse outcomes for patients such as an increased likelihood of institutionalization [3]. Various factors have been associated with CGB, albeit not consistently, such as cognitive function, stages of dementia, depression, activities of daily living (ADL), and behavior and psychological symptoms of dementia (BPSD) [4,5].

BPSD is considered one of the most difficult problems for caregivers to deal with [6] and is associated with higher levels of CGB, more rapid cognitive and functional decline, hospitalization, and institutionalization [7]. Risk factors for BPSD include various sociodemographic factors, disease severity, ADL, and health-related factors [6,7].

The impact of ADL and physical functioning on CGB have been mixed. Garre-Olmo [8] reported that BPSD and ADL had indirect effects on CGB via caregiver distress due to BPSD. Kim [6] found that ADL had indirect effects on CGB through various BPSD symptoms such as hyperactivity, psychosis, and physical behavioral symptoms, indicating that BPSD exerted a complex mediating effect. Sugimoto [7] found that physical vitality had direct and indirect effects, mediated through BPSD, on CGB. On the other hand, Onishi [9] found that physical disability did not affect CGB. The literature has yielded inconsistent findings between cognitive status and CGB [10], while some reports, such as those by Kim [6] and Sugimoto [7], found indirect effects through BPSD.

Until recently, most studies that looked at the relationships between CGB, BPSD, ADL, cognitive status, and physical illness used correlations and/or regression analysis. This led Kim and colleagues [6] to conclude that studies regarding the causal relationship between BPSD in community-dwelling patients with Alzheimer’s disease (AD) and caregiver burden “have not yet been established.” However, as noted above, several studies have employed path studies that allowed for assessing the complexity of the interactions, e.g., both the direct and indirect relationships between these variables. While these studies provided more insight into these relationships, there has been an increased recognition that other variables, particularly physical frailty, can further contribute to the understanding of the interactions that affect CGB and associated variables. Over the past decade, there has been growing interest concerning physical frailty in dementia and what impact it may have on CGB and BPSD.

Like dementia, frailty is thought to be among the most challenging public health issues of this century [11]. Physical frailty is common in AD, with rates ranging from 11% to 50% [12]. All older adults are at risk of developing frailty, and it is associated with many adverse outcomes including diminished quality of life and increased rates of mortality, hospitalizations, falls, depression, and dementia [11]. It is a dynamic process that is potentially preventable, reversible, and treatable. A recognition of frailty and its risk factors can inform treatment decisions and prognosis.

Physical frailty has been conceptualized as a “risk accumulation” model [13]—i.e., an accumulation of diseases and impairments that create a predisposition for adverse outcomes—or as a “syndrome” model [14], i.e., a set of signs and symptoms that define a health condition or phenotype. The accumulation model allows for the inclusion of disabilities and comorbidities such as dementia or cardiovascular disease, whereas the syndrome model is a “primary” or “preclinical” state that is not associated directly with a specific disease or disability.

Both BPSD and physical frailty were found to be significantly associated with each other [7] and with CGB in dementia [7,15,16]. However, only Sugimoto and colleagues’ [7] study in Japan has examined the interactions between physical frailty, caregiver burden, and BPSD in persons with Alzheimer’s disease. They found that frailty acted directly onCGB, as well as indirectly through BPSD; the latter also had an independent effect on CGB.

Race is another variable that has been overlooked in its relationship with the CGB and its associated factors. In the United States, race has been linked to a variety of disparities in health outcomes [17]. This has been especially true for dementia. Although empirical estimates suggest that the prevalence of AD in minority individuals is highly variable, the most conservative finding is that compared to non-Hispanic White individuals, Black/African American individuals are twice as likely, and Hispanic/Latino individuals are 1.5 times more likely to develop AD [18]. Race has been associated with CBG and physical frailty, with White individuals reporting a higher CGB but lower rates of frailty [2,19]. In dementia samples, frailty prevalence rates were nearly twice as high among Black and Hispanic individuals than among White individuals [19], and socioeconomic status and lower education exacerbated these racial differences [20]. In one study, racial differences tended to disappear after controlling for socioeconomic status [19], whereas another did not find this attenuation [20].

For caregivers in general, Black caregivers report a lower burden [2]. In dementia caregivers, the results are more mixed, ranging from no difference to less depression and more positive appraisals among Black caregivers [2]. There are also mixed findings for Hispanic caregivers, ranging from no differences compared to White caregivers, sometimes greater depression, and sometimes less perceived stress [2]. Greater racial differences were found in convenience samples. Factors contributing to this difference in CGB may include more positive perceptions of the caregiving role, greater religiosity, and more extended and supportive kin networks [2]. It must be underscored that although the caregiver burden may be perceived in a more favorable light, minority caregivers often experience substantial objective burdens, e.g., balancing work and caregiving, financial strains, and the accessibility and affordability of healthcare resources.

BPSD may be differentially expressed by race and is more likely to occur in Black participants with dementia than in White participants with dementia. In models adjusted for age, sex, and education, the odds of experiencing delusions and hallucinations were approximately doubled among Black individuals with dementia [21].

Several issues emerge from the review of the literature described above:Despite extensive research on CGB and BPSD, their causal relationships have not been fully established because of modeling that did not include potential confounding associated variables or examine the direct and indirect effects of variables on CGB and BPSD.There is increased recognition that physical frailty is common in dementia and may have direct and indirect effects on CGB and BPSD.In the United States, race has been found to impact CGB, frailty, and BPSD, but has not been systematically examined together with other variables that affect CGB, BPSD, and frailty such as ADL, cognition, and physical health.

To address these three issues, we use an exploratory path analysis to examine the relationships between CGB, BPSD, frailty, race, and other associated variables based on a model developed by Sugimoto’s team. In so doing, we provide guidance for larger confirmatory studies and discuss its implications for clinical care and research.

## 2. Materials and Methods

### 2.1. Study Population and Design

We used a cross-sectional design with data derived from the Alzheimer’s Disease Assistance Centers at four State University of New York campuses in various regions of the state (Buffalo, Syracuse, Brooklyn, Stony Brook). The institutional review boards at each site approved the research, and for this report, the SUNY Brooklyn IRB (no. 688786-7) provided approval. Study dyads (patient and caregiver) were recruited consecutively from new intakes at the four sites between 2014 and 2015. Patient inclusion criteria consisted of a primary diagnosis that met DSM-5 [22] criteria for a major neurocognitive disorder (dementia), a Mini-Mental State Examination(MMSE) score of 10 to 26 [23], a Clinical Dementia Rating Scale sum of boxes score of ≥3 [24], age ≥ 55, having a caregiver, and being English-speaking. We excluded major neurocognitive disorders caused by traumatic brain injury, HIV infections, Parkinson’s disease or Parkinson-related dementia, frontal dementia, substance abuse, or other medical causes. Of the 134 dyads screened, 85 met the study criteria. The sample was 67% female, had a mean age of 81.9 years (SD = 8.3), with a self-identified racial/ethnic distribution of 37% Black/African American, 58% White, 2% Hispanic/Latino, and 3% Other. Regarding living status, 22% lived alone, 34% with kin (child, sibling, or several relatives including spouse), 33% with a spouse alone, 9% with an unrelated caregiver, or 1% other. The primary caregiver was a spouse (58%), a female child (29%), a male child (8%), other kin (2%), or a friend (4%). Comprehensive evaluations of patients included physical, neuropsychiatric, and neuropsychological testing, along with ancillary blood work and neuroimaging. The initial assessments suggested that 78% (n = 66) of the sample had AD or AD with another neurocognitive disorder (“mixed” dementia), 8% (n = 7) had probable vascular dementia, 8% (n = 7) had possible Lewy Body Dementia, and 6% (n = 5) had mixed dementia other than with AD.

Our research design was based on an adaptation of Sugimoto and coinvestigators’ analytic model [7], as well as an incorporation of previous research on the relationship between race, caregiver burden, and frailty, described above. Although Sugimoto’s team examined only Alzheimer’s disease, many studies on caregiver burden have looked at dementia patients in general. We opted to focus on dementia patients to increase the power of our analysis and our concerns that many AD patients have mixed pathology [25]. However, a subanalysis of AD-diagnosed patients in our sample was also undertaken.

Within the overall model, caregiver burden was the dependent variable. Cognitive status, daily functioning, physical health, and race (White) were predictor variables, and BPSD was both a predictor and intervening variable between the latter variables and caregiver burden. In addition, SSF was conceptualized as a predictor variable of caregiver burden and BPSD, but also as an intervening variable for the relationship between race and caregiver burden. Cognitive status, daily functioning, and physical health were also conceptualized as having non-directional (symmetrical) relationships with SSF. Age and gender were used as covariates. This is depicted in Figure 1.

### 2.2. Variables and Instruments

To operationalize the model, we first addressed the assessment of physical frailty. As noted above, physical frailty has been conceptualized as a “risk accumulation” model [13]—i.e., an accumulation of diseases and impairments that create a predisposition for adverse outcomes—or as a syndrome model [14], i.e., a set of signs and symptoms that define a health condition or phenotype. The accumulation model allows for the inclusion of disabilities and comorbidities, whereas the syndrome model is a “primary” phenotypical state that is not associated directly with a specific disease or disability. To obviate the conflation of disability associated with dementia with that of frailty, we chose the syndrome (phenotype) model and separately examined impairments in activities of daily living and physical disorders. We used the SHARE Frailty Instrument (Share-FI) [26], which is a summed score ranging from 0 to 5, with scores of 1–2 and 3–5 classified as prefrail (subsyndromal) and frail (syndromal), respectively. We dichotomized the scores into nonfrailty versus subsyndromal/syndromal frailty (SSF). For daily functioning, we used Lawton’s Basic and Instrumental Activities of Daily Living (ADL) Scale [27] with a range of 0–14 (better); it was dichotomized using the median score of 7 as the cut point. To assess physical comorbidities, we used the Modified Cumulative Illness Rating Scale-Geriatrics (CIRSG) [28] that examines 14 medical systems and has a possible range of 0–56 (most severe); it was dichotomized using the median score of 7 as the cut point. The Neuropsychiatric Inventory (NPI) [29] was used to assess BPSD with a possible range of 0–144 (most frequent and severe symptoms). Caregiver burden was examined using the 4-item Zarit Caregiver Burden Interview (CBI) [30] with a possible range of 0–16 (most burdened). The MMSE (possible range in this study: 10–26) was used to assess cognitive status and was dichotomized into 17 and below (“severe”) and 18 and above (“moderate/mild”). Race was dichotomized into White and non-White. Age (dichotomized into 55 to 79; 80 and above) and gender (male/female) were used as covariates. The internal reliabilities (Cronbach’s alpha) of all scales were acceptable (≥0.74), except for the CIRSG, which was 0.56, or minimally acceptable [31]. Interviewers were trained using instructional sessions and videotapes.

### 2.3. Statistical Analysis

To test the model design, we used a path analysis that entailed three linear regression analyses with CBI, NPI, and SSF as the dependent variables, respectively. The former two regression analyses met assumptions of normality, and the latter met the criteria proposed by Hellevik [32] for using a dichotomous dependent variable in regression analysis. We used Wright’s method [33] to determine indirect effects; that is, we calculated the product of the betas (“compound correlations”) of the single paths comprising the multiple paths between independent and dependent variables. The Sobel test [34] was used to calculate the significance of these indirect effects. Because causal direction was predetermined in the path analysis, we used 1-tailed *p*-values with a significance level of *p* < 0.05. For associations between exogenous variables where the direction was not specified, 2-tailed *p*-values were used. The regression analyses were powered to detect small to medium effect sizes (f^2^ = 0.07). Any missing data, albeit rare, were replaced using mean imputation.

## 3. Results

Of the 59% (n = 50) of patients meeting the SSF criteria on the Share-FI, 46% (n = 23) scored 1–2 (“prefrail”/subsyndromal), and 54% (n = 27) scored 3–5 (“frail”/syndromal). Table 1 provides the mean values/percentages of the variables in the analysis. Table 2 provides the results of the three regression analyses. In Table 2, the unadjusted models show the bivariate relationships of the predictor variables with the dependent variables, and the adjusted model shows the independent (direct) effects of the variables when they are entered simultaneously into the analyses. No evidence of appreciable multicollinearity was found among the variables. The overall model predicting the CBI was significant [adjusted R^2^ = 0.24, F (8,76) = 4.36, *p* < 0.001]. Significant relationships (*p* < 0.05) within the various regression analyses were found for race, SSF, NPI, and CBI. Direct and indirect effects are shown in Figure 1. (The covariates, age and gender, were not included in the figure.) SSF had a significant indirect effect (compound correlation = 0.10) on the CBI mediated through the NPI (Sobel test = 1.77, *p* = 0.04). SSF did not have significant direct effects on the CBI (β = 0.09, *p* = 0.21), whereas the NPI had direct effects on the CBI (β = 0.42, *p* < 0.001). Race (White) had a direct effect on the CBI (β = 0.29, *p* = 0.003) and SSF (β = −0.19, *p* = 0.03); that is, race (White) was associated with a higher CBI and lower SSF. Race had no significant indirect effects mediated through SSF and the NPI (Sobel test= −0.67, *p* = 0.25) or SSF alone (Sobel Test = −0.76, *p* = 0.22). Neither the MMSE, CIRSG, or ADL scores were significant predictors of the CBI or the NPI, although ADL was significantly correlated with SSF (Figure 1). A post hoc analysis revealed that when the CIRSG vascular subscale was substituted for the CIRSG variable in the regression analyses, it had a significant relationship with the CBI (β = 0.26, *p* < 0.005) but not with the NPI (β = −0.18, *p* = 0.06).

The item of education had too many missing cases (n = 12) to be included in the primary analysis. Also, in the Sugimato model, it was not significant. However, we conducted a post hoc analysis that included education as a predictor variable in the analysis (n = 73) and found that education in the regression analyses was not significantly related to the CBI (β = 0.06, *p* = 0.30) or the NPI (β = 0.12, *p* = 0.18). It was significantly associated with SSF (r = −0.25, *p* = 0.04, 2-tailed) and race (White) (r = 0.33, *p* = 0.004, 2-tailed) and resulted in some attenuation of the relationship with the CBI in the case of White race (β = 0.15, *p* = 0.10) and with the NPI in the case of SSF (β = 0.20, *p* = 0.08).

Last, a subanalysis was performed with the patients diagnosed with AD (n = 66). As seen in Appendix A, all relationships among the variables remained the same, with no changes in significance for any of the variables.

## 4. Discussion

This study augments our understanding of the relationships between caregiver burden, physical frailty, race, BPSD, and other associated variables in persons with dementia. We partially confirmed Sugimoto and coinvestigators’ [7] model, in that SSF had indirect effects on the caregiver burden as measured by the CBI, the latter being mediated through the NPI, which in turn had direct effects on the CBI. Moreover, a significant association between frailty (SSF) and BPSD (measured by the NPI) was also confirmed. Unlike Sugimoto’s group, we did not find that SSF had direct effects on the CBI, although the beta in our study approximated the beta in their study. Also, consistent with the literature, White caregivers expressed higher levels of caregiver burden [2]. Conversely, being White was associated with significantly lower levels of SSF, an association that has been reported previously in non-dementia samples [19]. However, race had only direct effects on the CBI and did not have any significant indirect effects on the CBI through SSF or the NPI.

Our findings clarified some of the limitations of the Sugimoto team’s study. Because they used a frailty measure that was based on the accumulation model, it was difficult to determine whether frailty primarily reflected impairments in ADL and comorbid physical illnesses. Because we used the frailty phenotype construct and examined ADL and physical health separately, we demonstrated that SSF had significant indirect effects on the CBI, whereas ADL and CIRSG did not have any direct or indirect effects. Other studies have found indirect effects of ADL on caregiver burden mediated through BPSD [6]. Cheng [10] postulated that ADL may have a more profound effect on caregiver burden in advanced dementia, whereas, in persons with milder cognitive symptoms, such as in the sample reported here, ADL has less of an impact. Physical diseases, especially vascular disorders such as strokes, have been associated with more BPSD and greater caregiver burden [7,35,36,37]. We did not find this association for overall physical disorders, but a post hoc analysis looking at the vascular subscale of the CIRSG was significantly associated with greater caregiver burden. Future studies may be able to clarify these findings. Moreover, contrary to the Sugimoto group’s finding, we did not find that MMSE was associated with the NPI, although we did replicate their findings regarding the lack of an association between MMSE and caregiver burden. Indeed, the literature has yielded inconsistent findings between cognitive status and caregiver burden [10]. Finally, we were able to demonstrate the importance of race in this analysis, a variable that Sugimoto’s group did not include.

A key takeaway from this study and previous research is the pivotal role that BPSD plays as a mediator between various predictor variables and caregiver burden. This may have important implications for interventions to reduce the caregiver burden. In our study, it was an important mediator between frailty and caregiver burden, whereas in other studies, it mediated between the caregiver burden and ADL, physical health, or cognitive status [6,7,8]. Some investigators have examined various cluster types of BPSD symptoms and their mediating position between other variables and caregiver burden. However, there have been considerable differences in the results of the cluster analyses, so it has been difficult to draw any definitive conclusions [6].

A strength of this study is the sample’s multiracial composition and geographic diversity, as well as the fidelity of our analysis with the Sugimoto team’s design. To our knowledge, it is the first study to look at the impact of race, a critical social determinant of health outcomes in the United States, on the caregiver burden in dementia in concert with frailty and BPSD. Limitations include the cross-sectional design, meaning that causal direction cannot be verified; and the omission of potentially relevant variables in the model such as nutritional status and various social determinants such as social class and living circumstances. Moreover, although Sugimoto’s groups did not find education to be significant in their model, our findings that education may attenuate some of the effects of race on caregiver burden and the effect of frailty on the NPI suggests that education should be included in future research, especially when race is a predictor variable. Finally, because of the modest sample size (n = 85), there was a possibility of Type 2 errors, although the model was powered to detect small to medium effect sizes. Nonetheless, the findings must be viewed as provisional and need to be replicated longitudinally in other sites with larger sample sizes.

## 5. Conclusions

Our exploratory analysis demonstrated significant relationships between caregiver burden, frailty, race, and BPSD. The findings indicate that frailty, race, BPSD, and caregiver burden may directly or indirectly influence one another, and therefore should be considered essential elements of dementia assessment, care, and research.

## Figures and Tables

**Figure 1 medicina-60-00426-f001:**
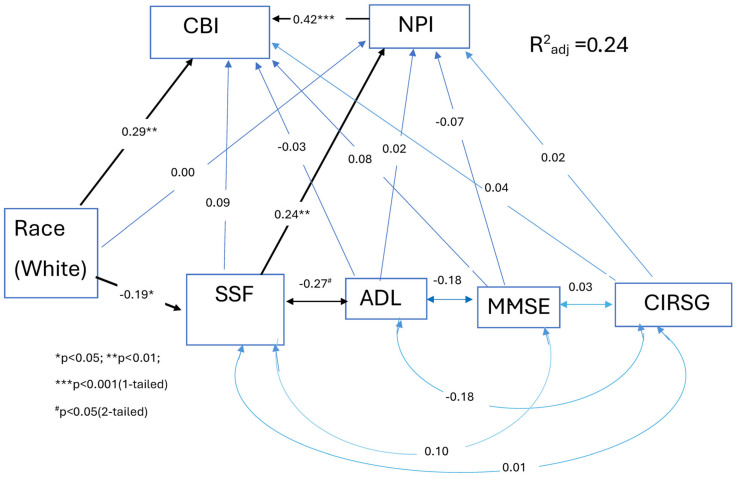
Path diagram representing the relationship between predictor variables. Abbreviations: SSF = subsyndromal/syndromal frailty; CBI = Caregiver Burden Interview; NPI = Neuropsychiatric Inventory; ADL= Activities of Daily Living Scale; MMSE= Mini-Mental State Examination; CIRSG = Cumulative Illness Rating Scale Geriatrics. Notes: 1. Indirect effect of SSF on CBI mediated by NPI: (0.24) (0.42) = 0.10. Sobel test: z-score = 1.77, *p* = 0.04 (1-tailed). 2. Indirect effect of race (White) on CBI mediated by SSF: (−0.19) (0.09) = −0.02. Sobel test: z-score = −0.76, *p* = 0.22 (1-tailed). 3. Indirect effect of race (White) on CBI mediated by SSF and NPI: (−0.19) (0.24) (0.42) = −0.02. Sobel test: z-score = −0.67, *p* = 0.25 (1-tailed).

**Table 1 medicina-60-00426-t001:** Demographic and clinical characteristics used in the analysis and inclusion criteria (N = 85).

Variable	Mean (SD)/%
Age	82.0 (8.3)
Gender (female)	67
Race (White)	58
Caregiver Burden Index	7.2 (3.9)
Neuropsychiatric Inventory	25.0 (21.6)
Activities of Daily Living Scale	7.0 (3.4)
Mini-Mental State Examination	19.2 (4.2)
Modified Cumulative Illness Rating Scale-Geriatrics	7.1 (3.7)
Clinical Dementia Rating Scale-Sum	7.4 (3.1)

**Table 2 medicina-60-00426-t002:** Linear regression analyses for path design.

Dependent Variables													
		CBI ^#^				NPI ^##^				SSF ^###^			
		Unadjusted Model		Adjusted Model		Unadjusted Model		Adjusted Model		Unadjusted Model		Adjusted Model	
Variables	Mean (SD)/%	β	*p*	β	*p*	β	*p*	β	*p*	β	*p*	β	*p*
Age > 80	71%	0.21	0.03	0.08	0.22	0.13	0.11	0.11	0.19	0.19	0.04	0.18	0.04
Female	67%	0.06	0.26	0.06	0.29	0.03	0.39	−0.05	0.34	0.33	0.001	0.27	0.005
White	58%	0.26	0.008	0.29	0.003	−0.04	0.38	0.00	0.49	−0.23	0.02	−0.19	0.03
MMSE < 17	34%	0.07	0.27	0.08	0.22	−0.04	0.36	−0.07	0.27	—	—	—	—
ADL > 7	47%	−0.18	0.05	−0.03	0.39	−0.06	0.29	0.02	0.43	—	—	—	—
NPI	25.0 (21.6)	0.44	<0.001	0.42	<0.001	—	—	—	—	—	—	—	—
CIRSG > 7	48%	0.13	0.12	0.04	0.37	0.01	0.48	0.02	0.43	—	—	—	—
SSF	59%	0.17	0.06	0.09	0.21	0.24	0.01	0.24	0.03	—	—	—	—

Notes: N = 85; the unadjusted model is the effect of each variable before entering all the variables simultaneously into the regression analysis (adjusted model); *p*-values are 1-tailed; ^#^ R^2^adj = 0.24, F(8,76) = 4.35, *p* < 0.001; ^##^ R^2^adj = −0.01, F(7,77) = 0.87 *p* = 0.53; ^###^ R^2^adj = 0.14, F(3,81) = 5.55, *p* = 0.002; Abbreviations: SSF = subsyndromal/syndromal frailty; CBI = Caregiver Burden Interview; MMSE = Mini-Mental State Examination; ADL = Activities of Daily Living Scale; NPI = Neuropsychiatric Inventory; CIRSG = Cumulative Illness Rating Scale Geriatrics.

## Data Availability

Upon request from the Institute for Healthcare Informatics, Jacobs School of Medicine and Biomedical Sciences, by contacting the first author, Carl I. Cohen: carl.cohen@downstate.edu.

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
