# Peer review of "The Relationships between Caregiver Burden, Physical Frailty, Race, Behavioral and Psychological Symptoms (BPSD), and Other Associated Variables: An Exploratory Study"

_medicina, 2024, doi:10.3390/medicina60030426_

Round 1

Reviewer 1 Report

Comments and Suggestions for Authors

Firstly, I would like to congratulate the authors for their dedication and work.

However, there are some points that should be addressed for the improvement of the

manuscript.

1) We suggest refraining from using the term “race” in the manuscript. Recent

anthropological studies have advised against the use of this word. Please see this article

for more details: “ We Should Abandon “Race” as a Biological Category in Biomedical

Research” https://www.ncbi.nlm.nih.gov/pmc/articles/PMC7682789/

INTRODUCTION

1) It would be interesting if the authors could provide more background information in

terms of which studies have already been published on the subject. We have found

additional articles that might be interesting to comment on the introduction:

a) Abreu W, Tolson D, Jackson GA, Costa N. A cross-sectional study of family

caregiver burden and psychological distress linked to frailty and functional

dependency of a relative with advanced dementia. Dementia. 2020;19(2):301-

318. doi:10.1177/1471301218773842

b) Ding, T.Y.G., De Roza, J.G., Chan, C.Y. et al. Factors associated with family

caregiver burden among frail older persons with multimorbidity. BMC

Geriatr 22, 160 (2022). https://doi.org/10.1186/s12877-022-02858-2

2) Please include the definition used for “physical frailty” and SSF and provide more

background information on BPSD

3) In the last paragraph of the introduction, authors must clearly state their research

hypothesis, what is new and original about their study, and the relevance of it.

For example, why studying SSF, what was the original hypothesis.

METHODS

1) The authors should clarify why they decided to conduct a pilot study. By

definition, a pilot study is used to assess the feasibility of an approach.

Considering that there are previous studies published on the same theme with

hundreds of participants, this study should be a larger scale study to add

significant information to the literature.

2) Please provide IRB number

3) Considering the authors have defined the study as a pilot study, no inferential

statistics should be proposed. We suggest consulting a professional statistician to

suggest the best statistical analysis for this pilot study.

RESULTS

A pilot study by definition does not test hypothesis, so authors should reconsider their

conclusions and results.

Comments on the Quality of English Language

Minor editing required

Author Response

Dear Editor:

Thank you for the opportunity to revise the paper. As suggested, we have considerably expanded the literature review in the introduction, and this gives readers a better appreciation of the rationale for the study. We have responded to all of the reviewers’ concerns below.

Response to Reviewer 1:

  1. We suggest refraining from using the term “race” in the manuscript. Recent anthropological studies have advised against the use of this word. Please see this article for more details: “ We Should Abandon “Race” as a Biological Category in Biomedical Research” https://www.ncbi.nlm.nih.gov/pmc/articles/PMC7682789/

One of the principal aims of this study is to examine the interaction of race in affecting caregiver burden, neuropsychiatric symptoms, and frailty. In this context, we use self-identified designations of race (see page 9). We are not using race as a biological category but as a socially constructed category that has implications for variables in the analysis.  The article cited above recognizes that the use of race as a socially constructed category is acceptable and to be differentiated from using it as an inherent biological category. As Smedley and Smedley (Am Psychol 2005;60:16–26) noted, “An abundance of evidence, however, demonstrates that race continues to matter in meaningful ways.”  Likewise, Braverman and Dominguez (Front Public Health. 2021 Sep 7;9:689462. doi: 10.3389) propose, “While not useful as a biological category, “race” as currently categorized—African American/Black, American Indian/Alaska Native, Asian, European American/White, Native Hawaiian/Pacific Islander, along with Latino/Hispanic “ethnicity”—is a vitally important social category for monitoring, understanding, and intervening on differences in health.”

  1. It would be interesting if the authors could provide more background information in terms of which studies have already been published on the subject. We have found additional articles that might be interesting to comment on the introduction: a) Abreu W, Tolson D, Jackson GA, Costa N. A cross-sectional study of family caregiver burden and psychological distress linked to frailty and functional dependency of a relative with advanced dementia. Dementia. 2020;19(2):301- 318. doi:10.1177/1471301218773842 b) Ding, T.Y.G., De Roza, J.G., Chan, C.Y. et al. Factors associated with family caregiver burden among frail older persons with multimorbidity. BMC Geriatr 22, 160 (2022). https://doi.org/10.1186/s12877-022-02858-2

In the introduction, we have considerably expanded our literature review. Abreu et al are included in the expanded review. We include more references on the interactions between caregiver burden, frailty, BPSD, race, and other associated variables. We focus on the impact of indirect effects on caregiver burden through various mediating variables, especially BPSD.

  1. Please include the definition used for “physical frailty” and SSF and provide more background information on BPSD

The major definitions of physical frailty are now defined in the introduction on pp 5-6.

  1. In the last paragraph of the introduction, authors must clearly state their research hypothesis, what is new and original about their study, and the relevance of it. For example, why studying SSF, what was the original hypothesis.

After our expanded literature review in the introduction, we now indicate three unresolved issues that emerged from the review, and how we plan to address these issues using an exploratory path analysis to examine the relationships between CGB, BPSD, frailty, race, and other associated variables based on a model developed by Sugimoto’s team ( see pp7-8). In so doing, we aim to provide guidance for larger confirmatory studies as well as discuss its implications for clinical care and research. We believe this now provides a much stronger rationale for our study and its methods.

  1. The authors should clarify why they decided to conduct a pilot study. By

definition, a pilot study is used to assess the feasibility of an approach.

Considering that there are previous studies published on the same theme with

hundreds of participants, this study should be a larger scale study to add

significant information to the literature.

In various drafts of the papers, we considered using “exploratory study’ instead of “pilot study.” As the reviewer is undoubtedly aware, this is no consensus regarding these definitions.  There are many definitions of “pilot study,“ and some definitions include data analysis and hypothesis testing, and some writers view it as a full research study, but on a smaller scale, the results of which may affect the design of a larger study. For example, Hallingberg and colleagues’ (Pilot and Feasibility Studies (2018) 4:104) literature review of this topic stated for inclusion, “ A study which aims to generate the evidence needed to decide whether and how to proceed with a full-scale effectiveness trial, or other study design and are labelled as exploratory/pilot/feasibility/ phase II/proof of concept. Eligible publications may concern some or all of the design features of exploratory studies.”  On the other hand, we believe that the differentiation of “pilot” from “exploratory” study described in a Mayo Clinic presentation is consistent with the reviewer’s definition. ( See Karla Ballman “ Pilot Study Design Testing https://player.slideplayer.com/18/6159163/# ) She proposes that pilot studies look at “objective something’ other than hypothesis generation, that it is a prelude to a larger study, and there is not an explicit justification of the sample size. Alternatively, she defines “exploratory” studies as hypothesis-generating, standing on its own, and there is an explicit justification for the sample size. Based on her definition, and more in line with the reviewer’s comment, we have revised the title of our study to “exploratory” since it is more consistent with our intent. That is, we are expanding and testing a model proposed in the literature but on a smaller scale with the aim that it can provide a basis for a larger study. Moreover, we have specified a sample size that can detect between small and medium effect sizes.

  1. Please provide IRB number

IRB no. 688786-7.  Inserted in the text on page 8. It was also included in the section after the references.

  1. Considering the authors have defined the study as a pilot study, no inferential

statistics should be proposed. We suggest consulting a professional statistician to

suggest the best statistical analysis for this pilot study.

As noted above, consistent with our intent, we have retitled the paper as an “exploratory study” and believe this recategorization allows for model testing. Concerning the statistical methods. I have extensive experience in statistics but also consulted a long-time collaborator, Robert Yaffee, Ph.D., an experienced statistician who worked for many years at NYU and authored numerous papers and books.

It should be noted there are two acceptable analytic strategies: Path Analysis and Structural Equation Modeling. The latter is an extension of path analysis that uses latent variables to account for measurement errors. Path analysis assumes all variables are measurable without errors.  Both try to detect relations among variables in a model (direct and indirect effects). Although SEM is potentially more rigorous, path analysis in an exploratory study such as this one is acceptable. Onishi et al (2005), cited in our paper, used this strategy. Moreover, because we employed established scales, the confirmatory factor analysis of SEM for creating new scales is not essential. Finally, a critical limitation to using SEM is our sample size. Most statisticians would consider our sample size to be too small. Sample sizes of at least 100 to 150, or even 200 are recommended. See: https://www.oreilly.com/library/view/structural-equation-modeling/9781118356302/c07anchor-1.html

8.. A pilot study by definition does not test hypothesis, so authors should reconsider their conclusions and results.

As noted above, based on our recategorization of the study as “exploratory,” we believe the definition of exploratory study is consistent with testing of the Sugimoto model.   

Reviewer 2 Report

Comments and Suggestions for Authors

The authors investigated the relationship between physical frailty, race, BPSD, and caregiver burden in persons with dementia. Previous study reported the relationship between physical frailty, BPSD, and caregiver burden, without race, because the study was conducted in Japan. Most of the population in Japan is made up of Asians. The authors conducted a similar analysis with a new focus on race. Race (White) had significant direct effects on frailty and caregiver burden.

What is the reason for the differences between races? Is there a relationship between educational level and economic environment? There is no medical usefulness unless this point is discussed.

This paper is written like Data mining analysis. It is better to focus on the relationship between race and other factors when describing the results to distinguish them from previous studies.

The summary of Sugimoto et al.'s study seems to be included in the results section, but it should be summarized in the introduction session.

Participants included vascular dementia (15%) and possible Lewy body dementia (12%). It will also be necessary to describe the results for Alzheimer's disease alone because symptoms vary considerably depending on the underlying diseases.

A path diagram will be included in the results section.

Author Response

Reviewer 2

  1. What is the reason for the differences between races? Is there a relationship between educational level and economic environment? There is no medical usefulness unless this point is discussed.

 We have now cited several articles from the literature that have addressed some of these issues of education and SES (see pp 6-7).  Although we had no information on the economic environment (income items were often refused), we had some educational information, although there was some missing data. We now describe these findings as a posthoc analysis (page 16) as follows:

“The item on education had too many missing cases (n=12) to be included in the primary analysis. Also, in the Sugimato model, it was not significant. However, we conducted a post-hoc analysis that included education as a predictor variable in the analysis (n=73) and found that education in the regression analyses was not significantly related to CBI (β=.06, p=.30) or the NPI (β=.12, p=.18). It was significantly associated with frailty (r=-.25, p=.04, 2-tailed) and race(white) (r=.33, p=.004, 2-tailed) and resulted in some attenuation of the relationship with the CBI in the case of white race (β=.15, p=.10) and with the NPI in the case of frailty (β=..20,p=.08).”

This is further examined in the discussion section (page 21):

“Moreover, although Sugimoto’s groups did not find education to be significant in their model, our findings that education may attenuate  some of the effects of race on caregiver burden and the effect of frailty on the NPI suggests that education should be included in future research, especially when race is a predictor variable.”

  1. This paper is written like Data mining analysis. It is better to focus on the relationship between race and other factors when describing the results to distinguish them from previous studies.

 On the contrary, we aimed to avoid data mining. We used Sugimoto’s model so that we specified specific variables to be examined in advance rather than randomly looking for significant variables. Moreover, we now have provided an expanded literature review in the introduction that provides additional justification for the inclusion of the variables in the analysis. Indeed, the study’s main contribution is to look at direct and indirect relationships among caregiver burden, race, frailty, BPSD, and other associated variables using a path model. Looking at race alone would not accomplish this goal.

  1. The summary of Sugimoto et al.'s study seems to be included in the results section, but it should be summarized in the introduction session.

We now have moved this to the introduction section (page 6).

  1. Participants included vascular dementia (15%) and possible Lewy body dementia (12%). It will also be necessary to describe the results for Alzheimer's disease alone because symptoms vary considerably depending on the underlying diseases.

 We have now examined a subsample of patients diagnosed with AD and ran the regression analyses. The results were very similar. We note this in the text on page 16 and include these data in a supplementary table.

Round 2

Reviewer 1 Report

Comments and Suggestions for Authors

Satisfactory